# COMPASS: joint copy number and mutation phylogeny reconstruction from amplicon single-cell sequencing data

Etienne Sollier [1,2], Jack Kuipers [1,3], Koichi Takahashi [4,5], Niko Beerenwinkel [1,3] & Katharina Jahn [1,3,6] ✉

Reconstructing the history of somatic DNA alterations can help understand the evolution of a tumor and predict its resistance to treatment. Single-cell DNA sequencing (scDNAseq) can be used to investigate clonal heterogeneity and to inform phylogeny reconstruction. However, most existing phylogenetic methods for scDNAseq data are designed either for single nucleotide variants (SNVs) or for large copy number alterations (CNAs), or are not applicable to targeted sequencing. Here, we develop COMPASS, a computational method for inferring the joint phylogeny of SNVs and CNAs from targeted scDNAseq data. We evaluate COMPASS on simulated data and apply it to several datasets including a cohort of 123 patients with acute myeloid leukemia. COMPASS detected clonal CNAs that could be orthogonally validated with bulk data, in addition to subclonal ones that require single-cell resolution, some of which point toward convergent evolution.

Intratumour heterogeneity plays a key role in the failure of targeted cancer therapies[1]. Obtaining a comprehensive picture of the clonal architecture and the mutational history of a patient's tumour at the timepoint of diagnosis therefore offers great potential to improve treatment choices and predict disease progression. Single-cell DNA sequencing (scDNAseq) generally provides a higher resolution of intratumour heterogeneity than sequencing bulk tumour samples. However, this advancement comes at the cost of higher levels of noise primarily introduced during DNA amplification, an essential preparatory step for scDNAseq. As tumours typically evolve through a combination of single-nucleotide variants (SNVs) and copy number alterations (CNAs), it has been a critical limitation that current DNA amplification technologies do not permit the reliable calling of SNVs and CNAs simultaneously from the same cells. Multiple displacement amplification (MDA)[2], which is used in most scDNAseq protocols, provides a high coverage and has a low error rate and is therefore well suited to detect SNVs. However, MDA results in amplification biases, which preclude reliable detection of CNAs[3]. Other protocols are better

suited to detect CNAs but not SNVs, for example a shallow whole-genome sequencing (WGS) as introduced by 10x Genomics[4]. Recently, a high-throughput microfluidics approach was introduced, which processes thousands of single cells while sequencing only a small set of disease-specific genes[5], and later commercialised by Mission Bio, Inc. as the Tapestri® platform. While the limited physical coverage of the genome is far from ideal for calling copy number events, which can stretch anywhere from a small number of bases to whole chromosomes, this approach allows for the use of targeted PCR in the amplification step which does not introduce the strong amplification biases observed in MDA and therefore allows, in principle, to infer both SNVs and CNAs from the same cells[6].

Method development for inferring the evolutionary history of tumours from scDNAseq data closely followed the technology development (Table 1). Initially, approaches have been developed to reconstruct SNV-based mutation histories[7–13]. Later methods were introduced that analyse the history of copy number variants[14,15]. SCARLET[16] was the first method for single-cell data that tried to bridge

[1]Department of Biosystems Science and Engineering, ETH Zürich, Basel, Switzerland. [2]Division of Cancer Epigenomics, German Cancer Research Center (DKFZ), Heidelberg, Germany. [3]SIB Swiss Institute of Bioinformatics, Basel, Switzerland. [4]Department of Leukemia, The University of Texas MD Anderson Cancer Center, Houston, TX, USA. [5]Department of Genomic Medicine, The University of Texas MD Anderson Cancer Center, Houston, TX, USA. [6]Department of Mathematics and Computer Science, Freie Universität Berlin, Berlin, Germany. ✉e-mail: katharina.jahn@fu-berlin.de

**Table 1 | List of methods for tumour phylogeny inference from scDNAseq data, with their main features**

| Method | SNVs | CNAs | Doublets | SNV Recurrence | SNV loss | Homozygous mutations | Est. max # cells | Est. max # loci |
|---|---|---|---|---|---|---|---|---|
| ∞SCITE[7, 8] | Yes | No | Yes | Yes[a] | Yes[a] | No | 10,000 | 100 |
| SCIΦN[9, 10] | Yes | No | No | Yes | Yes | No | 100 | 1000 |
| OncoNEM[11] | Yes | No | No | No | No | No | 100 | 100 |
| SiCloneFit[12] | Yes | No | Yes | Yes | Yes | No | 100 | 100 |
| SPhyr[13] | Yes | No | No | No | Yes | No | 100 | 100 |
| SCICoNE[14] | No | Yes | No | – | – | – | 100 | – |
| CHISEL[15] | Yes[b] | Yes | No | – | – | – | 1000 | – |
| SCARLET[16] | Yes | No[c] | No | No | Yes[d] | No | 100 | 100 |
| BiTSC2[17] | Yes | Yes[e] | No | No | Yes | Yes | 100 | 100 |
| COMPASS | Yes | Yes | Yes | No | Yes[f] | Yes[f] | 10,000 | 100 |

The maximum number of cells and loci are estimates for reasonable runtimes and performance.
[a]However model selection is not automated.
[b]Can assign SNVs to clones after the CNA-tree is inferred by aggregating all cells assigned to each clone.
[c]Requires CNA tree as input, which must be obtained with another method.
[d]If supported by copy-number loss; which could miss CNLOH.
[e]Assumes that all loci have the same coverage (in the absence of CNAs), which is not the case for targeted sequencing.
[f]With copy number loss or CNLOH.

the gap between SNV- and CNA-based tumour phylogeny reconstruction. It infers an SNV phylogeny with CNA-constrained loss of heterozygosity (LOH), but the CNA tree has to be obtained separately, from different cells of the same tumour. BiTSC2[17] is the only existing method that can jointly infer the phylogeny of SNVs and CNAs. Its main drawback is that it assumes that in the absence of copy number events, the coverage is uniform across the genome, which, in our experience, is not the case for amplicon sequencing data. BiTSC2 also does not model copy number-neutral loss of heterozygosity (CNLOH), and might therefore falsely interpret such events as copy number losses, and does not scale well to a large number of cells.

Here, we introduce COMPASS (COpy number and Mutation Phylogeny from Amplicon Single-cell Sequencing), a probabilistic model and inference algorithm that can reconstruct the joint phylogeny of SNVs and CNAs from single-cell amplicon sequencing data. Its key features are that it models amplicon-specific coverage fluctuations and that it can efficiently process high-throughput data of thousands of cells. We show in simulation studies that COMPASS vastly outperforms BiTSC2 in settings where coverage variability resembles targeted scDNAseq. On data with uniform coverage, both methods perform very well with a slight advantage for COMPASS in most settings. We apply COMPASS to three datasets: a large cohort of 123 patients with acute myeloid leukemia (AML)[18], 4 TP53-mutated AML patients before and after venetoclax treatment[19] and 8 TP53-mutated myeloproliferative neoplasms (MPN)[20]. Furthermore, we orthogonally validate our findings with bulk sequencing and SNP array data.

## Results
### Probabilistic model for joint SNV and CNA single-cell tumour phylogenies
We have developed COMPASS, a likelihood-based approach to infer the evolutionary tree of somatic events in a tumour from single-cell panel sequencing data. The set of somatic events considered by COMPASS comprises SNVs and CNAs: gains, losses and CNLOH (Fig. 1A). CNAs affect regions and for panel-sequencing data we considered as regions genes, by grouping together amplicons targeting the same gene. One region may contain no variant (like region B in the example of Fig. 1C), one variant (region C) or several variants (region A). When variants are present in a region, the CNA calls are allele-specific. SNVs are acquired only once, while one region can be affected by several CNAs, but at most once per lineage (Fig. 1B). We limited the number of CNAs to at most once per lineage because the noisiness of single-cell data makes it difficult to infer exact copy numbers beyond

three, and because a loss followed by a gain (or vice-versa) in the same region would be difficult to detect with targeted sequencing data. An SNV can be lost multiple times in different lineages, which correspond to the Dollo model[21]. Germline SNPs can also be included in addition to somatic SNVs to improve the CNA inference (blue SNV in Fig. 1C). When this is done, COMPASS will automatically detect that these variants are present in the non-neoplastic cells and will place them at the root of the tree.

COMPASS uses as input the reference and mutated read counts, for each variant in each cell, and the number of reads covering each region (Fig. 1C). The total number of reads in a region is used by the probabilistic model to infer copy-number gains and losses (Fig. 1E) and the variant read count is used to detect SNVs and is also taken into account for CNAs which lead to an allelic imbalance (Fig. 1D).

In a tree of somatic evolutionary events, each node implies a genotype, which is obtained by altering the wild-type diploid genome by the sequence of events defined by the path from the root to the node. By assigning cells to a genotype associated with a tree node, the likelihood of the observed cell-specific read count profiles can be computed, as is described in the methods section. In order to compute the likelihood of the tree of somatic events, COMPASS marginalises out the assignment of cells to node genotypes, which is much more computationally efficient than sampling the attachments of cells to nodes when the number of cells is high. To account for the major sources of noise in scDNAseq data, COMPASS models sequencing errors, allele-specific dropout rates, and doublets. For tree inference, we define a prior distribution on trees that penalises the number of nodes and of CNAs to explain the observed sequencing data. A simulated annealing algorithm is then used to infer the tree that maximises the posterior probability.

### Evaluation on synthetic data
We evaluated COMPASS on synthetic data and compared it against BiTSC2[17], which is the only other method that can infer a joint SNV- and CNA-based tumour phylogeny. We also included SCITE[7], an established method of SNV-based tumour phylogeny, in order to highlight the benefits of joint SNV and CNA inference over SNV-only inference. We generated data that resembles data produced by the Tapestri® platform, as described in Supplementary Note 4.1. We used 3000 cells, 30 regions and trees with 6 nodes and different numbers of SNVs and CNAs. Surprisingly, we noticed that BiTSC2 performed worse with a large number of cells (Supplementary Fig. 12), in addition to having a very long runtime (Fig. 2F). To accommodate for this, we subsampled BiTSC2's input to 200 cells. The Tapestri® platform produces data

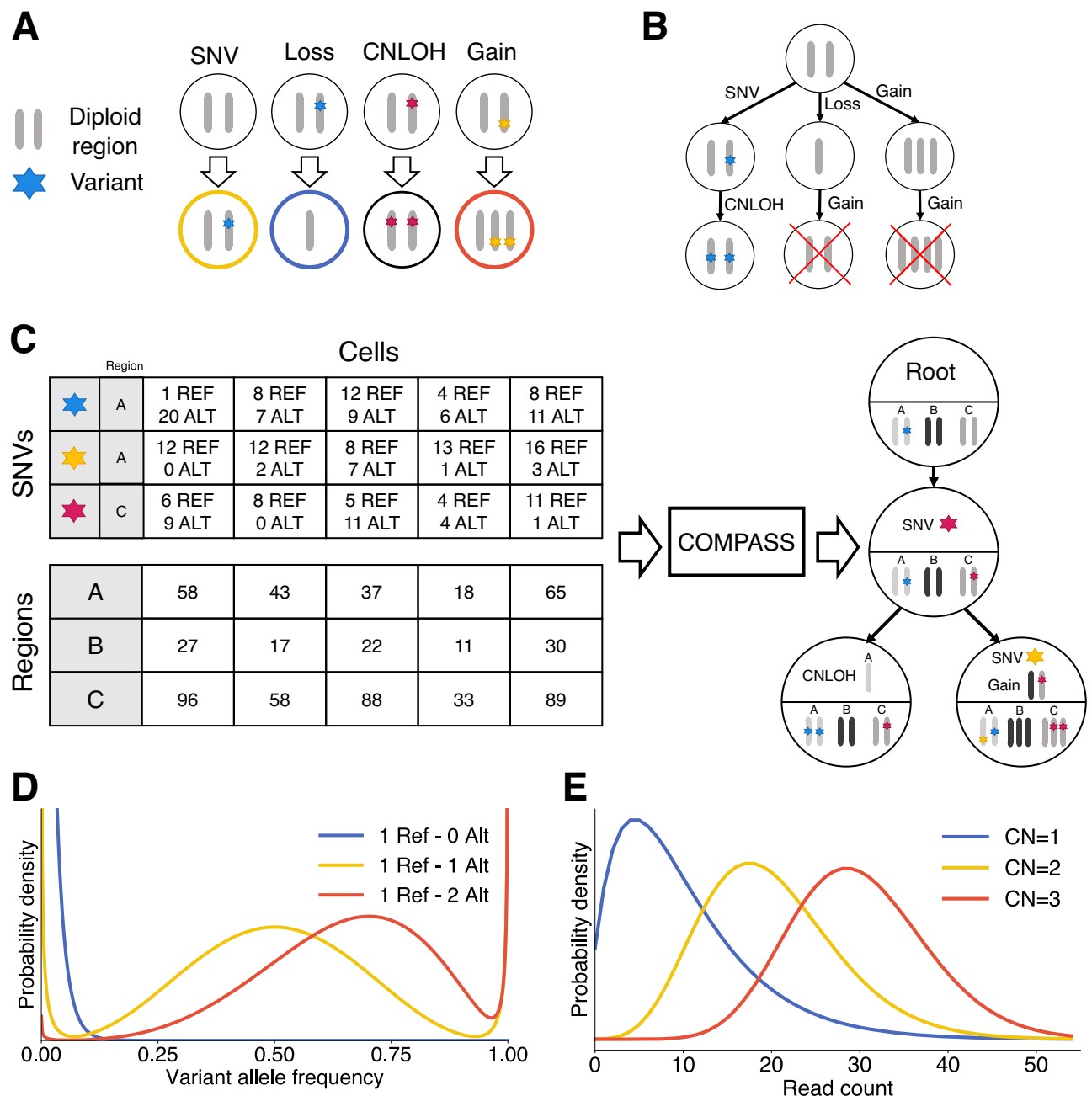

**Fig. 1 | Overview of COMPASS. A** Four types of somatic events included in the mutation tree. The CNAs (loss, gain and CNLOH) are allele-specific and can affect any of the two alleles, provided that the affected allele is already present in the node. **B** SNVs are acquired exactly once and CNAs can affect the same region multiple times, but at most once per lineage. **C** Input and output of the COMPASS algorithm. COMPASS needs the number of reads in each region in each cell and the number of reads supporting the reference (REF) and alternative (ALT) allele for each variant. COMPASS infers a tree of somatic events (SNVs and CNAs). These somatic events imply a genotype for each node, which is depicted in the lower part of each node. **D** Likelihood of the variant allele frequency for different copy numbers of the reference and alternative alleles (here for a fixed total number of reads covering the locus), based on a beta-binomial distribution with dropouts. **E** Likelihood of the total read counts in a region depending on the copy number of that region (here for a mean sequencing depth of 20 for a diploid region), which is based on a negative binomial distribution.

where the coverage is not uniform across amplicons, since each pair of primers has its own efficiency (Supplementary Figs. 6–7). In order to evaluate the impact of this uneven coverage on the performance of the different methods, we generated data with uniform and non-uniform coverage across amplicons. We evaluated the performance by MP3 similarity[22] between the inferred and the true tree. The MP3 similarity is defined on mutation trees where each node contains a set of mutations, and can be applied to trees which do not have exactly the same set of mutations. Here, we assigned a unique label to each SNV and to each CNA (defined by the affected region and whether the CNA is a gain or a loss), such that the MP3 similarity captures the correctness of both the detected CNAs and the inferred tree topology. In order to better understand the impact of SNVs and CNAs on the MP3 similarity, we also computed the MP3 similarity where we included only SNVs or only CNAs in the trees. Furthermore, we evaluated the correct assignments of cells to nodes, by first mapping each node of the inferred tree to the node of the true tree with the most similar genotype.

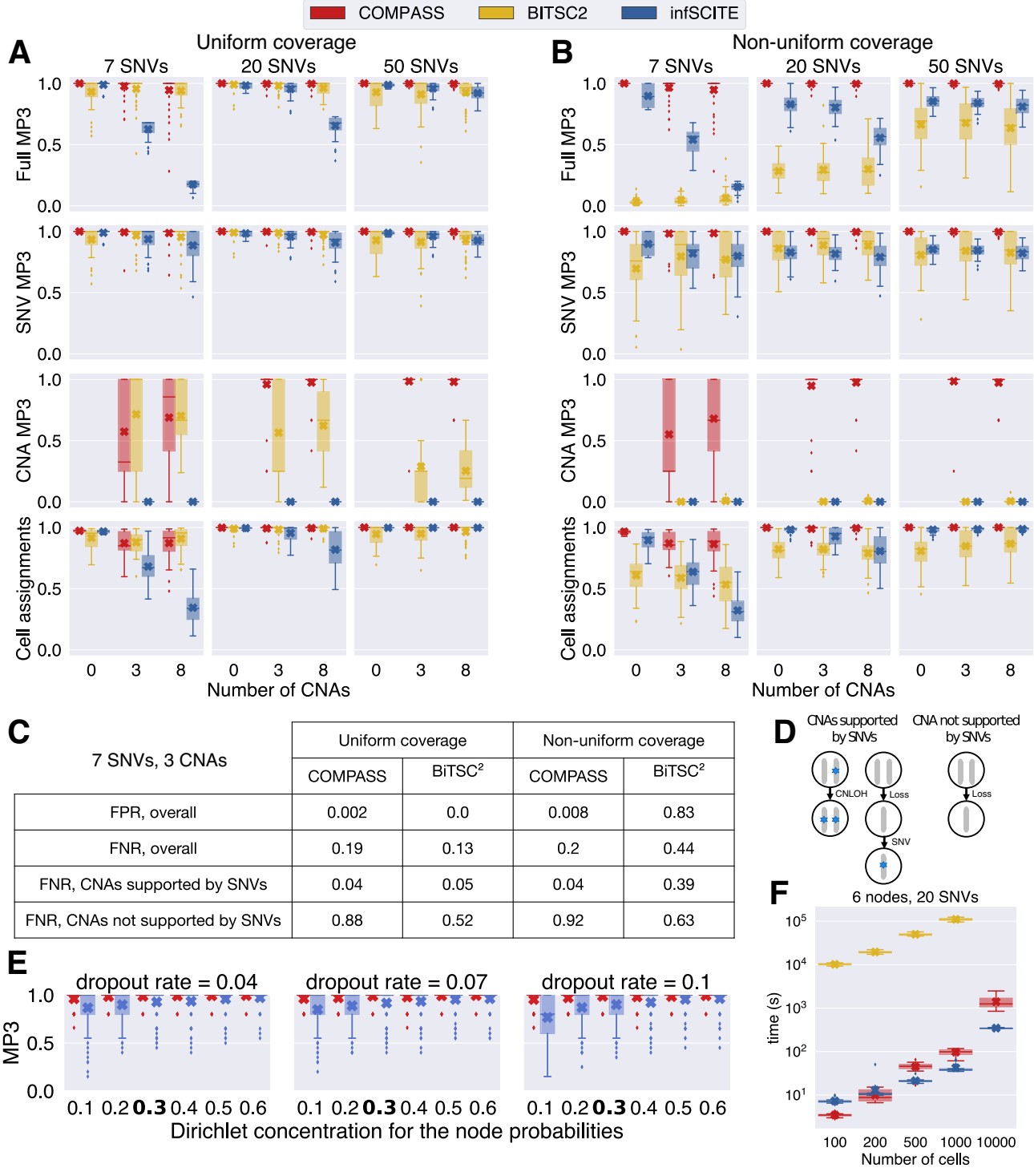

**Fig. 2 | Evaluation of COMPASS, BiTSC² and SCITE on synthetic data.** The box-plots represent the median and first and third quartiles, crosses indicate means, whiskers show the rest of the distribution up to 1.5 times the interquartile length, and outliers not within this range are shown as diamonds. **A, B** MP3 similarity (full, only SNVs or only CNAs) between the inferred and true trees, and accuracy of cell assignments, for different number of SNVs and CNAs, with uniform (**A**) or non-uniform (**B**) coverage. For each setting, we generated 50 different trees. **C** False positive rate (FPR) and false negative rate (FNR) for the CNA calls of COMPASS and BiTSC², when 7 SNVs and 3 CNAs were used. **D** Sketch explaining our definition of CNAs supported by SNVs: these CNAs contain in the same node, or in one of their descendant, a SNV or a LOH. **E** Evaluation of SCITE with different dropout rates and concentration parameters for the node probabilities (the higher the concentration parameter is, the more similar the node sizes are). The highlighted value (0.3) corresponds to the one estimated from real data. **F** Runtimes.

COMPASS was found to perform best in all settings we analysed (Fig. 2A, B). BiTSC²'s performance was close to COMPASS when the coverage was uniform, but its performance dropped sharply when the coverage was non-uniform. BiTSC² assumes that in the absence of CNAs, all loci have the same coverage, which is not a valid assumption for targeted sequencing. In addition, BiTSC² does not model CNLOH events. Although COMPASS performed very well, we observed that its performance for CNA inference was lower when the number of SNVs was lower. As described in the "Methods" section, COMPASS first infers the best tree without gains and losses, identifies regions whose

coverage at one node differs from their coverage at the root, and selects those regions as candidate regions that might harbour a gain or a loss. COMPASS then looks for the best tree, but allowing only gains and losses in these selected regions. This approach drastically reduces the number of false positive CNA calls, but decreases the sensitivity to detect subclonal CNAs. If a CNA is located in a subclone that contains an SNV or LOH event, the subclone will be present in the tree without gains and losses, and the corresponding region should be selected, enabling the detection of this CNA. However, if a subclone is only defined by a gain or loss which does not result in a LOH, it will be missing from the tree without gains and losses, and the CNA will not be detected. To quantify this phenomenon, we say that a CNA is supported by SNVs if the CNA is in a node that contains an SNV or a LOH, or that has a descendant containing such an event (Fig. 2D). As expected, the false negative rate of COMPASS for CNAs not supported by SNVs is high (Fig. 2C), but for CNAs supported by SNVs it is much lower than that of BITSC[2] with non-uniform coverage. The decreased ability of COMPASS to detect CNAs in subclones not supported by SNVs is counterbalanced by a very low false positive rate. In contrast, BiTSC[2] is significantly less conservative in calling CNAs when the coverage is not uniform, as indicated by its very high false positive rate.

SCITE performed well in the absence of CNAs, but its performance dropped when CNAs were included, which is expected since it does not model CNAs. Interestingly, the MP3 similarity based only on SNVs was already negatively affected by the presence of CNAs, highlighting the benefit of a joint inference. Even in the absence of CNAs, we observed that COMPASS slightly outperformed SCITE. As described in the methods section, COMPASS computes the likelihood of a tree by assigning a prior attachment probability to each node, whereas SCITE uses the same probability for each node, which is not a valid assumption when different clones have different sizes. We generated trees with different values for the Dirichlet concentration parameter for the node probabilities (high values: all nodes have the same number of cells attached to them; low values: variable number of cells for each node) and without CNAs. We observed that SCITE performed worse when nodes have different numbers of cells attached to them, including for the typical value observed in real data, and this effect was more pronounced when the dropout rate was higher (Fig. 2E).

All methods were found to be very robust to the presence of doublets in the data (Supplementary Fig. 13). Only when the doublet rate is very high does the performance drop, and this can be alleviated by using the models of COMPASS or SCITE which explicitly account for doublets (at the cost of an increased computational time).

**Correlations between the coverage at different amplicons**
When there are no CNAs, we would expect the sequencing depth on each amplicon to be independent. However, we observed strong correlations between the relative sequencing depth on different amplicons (Supplementary Fig. 9). Such correlations in Tapestri® data have not been reported before. The biological explanation for these correlations is not clear, but they have the potential to confound the CNA inference, since we could interpret the two main clusters as two different clones with very different copy number profiles. However, these correlations are independent from the actual clonal architecture of the tumour, so by jointly inferring SNVs and CNAs, only the true CNAs should be detected. We simulated data with such correlations between the coverage of different regions, and verified that these correlations did not affect the results of our method (Supplementary Fig. 14).

**Overview of CNAs detected in real AML data**
We applied COMPASS to a cohort of 123 AML patients that were previously profiled with the Tapestri® platform[18]. These samples were sequenced using two different panels: 67 samples with a 50-amplicon panel covering 19 genes and 53 samples with a 279-amplicon panel

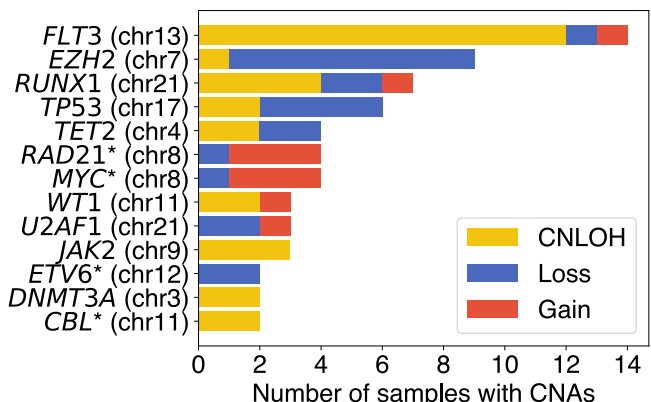

**Fig. 3 | Number of CNAs detected by COMPASS on each gene in the cohort of 123 AML patients.** Asterisks indicate genes covered only by the larger panel.

covering 37 genes (the genes covered by the panels are listed in Supplementary Table 1). In total, COMPASS detected CNAs in 42 samples (Supplementary Table 2): 31 CNLOHs, 26 deletions, 12 gains. The most common CNAs detected by COMPASS correspond to alterations which are known to be common in AML (Fig. 3). For example, the most common CNA detected by COMPASS in this cohort was CNLOH of *FLT3* (N = 12 samples, 9.8%), and *FLT3* is indeed known to be commonly affected by CNLOH in AML, especially when there is an internal tandem duplication[23,24]. Likewise, COMPASS detected many deletions of *EZH2* (N = 8 samples, 6.5%), and deletion of the long arm of chromosome 7 is indeed a common alteration in AML[25]. Interestingly, all samples with the *JAK2* p.V617F mutation also had a CNLOH, and this often occurred in a very small subclone (Supplementary Fig. 20), in agreement with previous reports of CNLOH for this mutation[26]. In AML, mutations in the *TP53* gene are known to be associated with a complex karyotype (at least three cytogenetic alterations)[27]. We tested if we could also find such an association based on our analysis. Reassuringly, we indeed detected mutations in *TP53* in 9 samples, 5 of which also had at least one gain or loss (p = 0.009, Fisher's exact test, one-sided, oddsratio = 6.96, 95%CI = [1.7; +∞), with 17 samples with gains and losses without *TP53* mutation and 97 samples with neither). We note that the ability of COMPASS to detect CNAs depend on how many regions are covered by the targeted panel used. Not surprisingly, more CNAs were detected in samples analysed with the 279-amplicon panel than with the smaller 50-amplicon panel. For example, we found 4 samples with CNAs on chromosome 8 with the larger panel, but the smaller panel does not contain any amplicon on this chromosome. The long arm of chromosome 5 is frequently deleted in AML, but in both panels there was only one amplicon targeting it (on *NPM1*) which had a low coverage, and could therefore not be used for CNA inference.

**Orthogonal validation of COMPASS-derived CNA calls with bulk data**
Bulk-targeted sequencing covering 297 genes was available for 85 out of 123 samples. We used CNVkit[28] to detect CNAs (only gains and losses, no CNLOH) in these samples. In addition, we had SNP array data available for 32 samples, for which we used ASCAT[29] to detect CNAs (including CNLOH). These bulk data provide an opportunity to orthogonally validate the CNA calls of COMPASS with more established (but lower resolution) approaches. We restricted the validation to events present in more than 50% of the cells, since bulk data cannot reliably detect CNAs present in a small percentage of the cells.

Among the 17 samples for which we detected gains or losses, bulk SNP array data was available for 2 of them and bulk targeted sequencing for 6 of them. Among these 8 samples, all of the CNAs present in a majority of the cells identified by COMPASS were also detected by bulk

sequencing, except for a gain affecting *ASXL1* in sample AML-60-001 (Fig. 4 and Supplementary Fig. 15). In the 85 samples with bulk sequencing, only one contained CNAs detected in bulk data on regions covered by the single-cell panel that were not detected by COMPASS (trisomy 8 for sample AML-81-001, Supplementary Fig. 16).

Among the 32 samples for which reliable SNP array data was available, six of them contained CNLOH events detected by both COMPASS and ASCAT, one sample contained a CNLOH event detected by COMPASS but not ASCAT, and four samples contained CNLOH events detected by ASCAT but not COMPASS, but those were either in regions not targeted by any amplicons or where we did not detect any SNVs (Supplementary Fig. 17).

For sample AML-59-001, COMPASS inferred a tree containing two main clones, each of which has a different mutation in the *RUNX1* gene (Fig. 4B). In addition, the dominant clone has one deletion of *EZH2* on chromosome 7, and one amplification of *WT1* on chromosome 11, while the smaller clone has a loss of *TP53* on chromosome 17, which results in a LOH for one germline variant (the sample might also have a somatic mutation on *TP53* not captured by the panel). The ASCAT profile inferred from SNP array data also contains the deletion on chromosome 7 and the amplification of chromosome 11, but does not contain any loss on chromosome 17 (Fig. 4A). This is expected, as this deletion is only present in 5% of the cells and hence cannot be detected from a bulk sample. This example supports the correctness of our tree since the CNAs found in the dominant clone are also detected with an orthogonal method. We investigated the plausibility of the CNAs inferred by COMPASS further by plotting the coverage on the three genes affected by CNAs depending on the cells' genotypes (Fig. 4E). The cells with the *RUNX1* p.D198N mutation do indeed have a lower coverage on *TP53*, while those with the *RUNX1* p.N153fs have a lower coverage on *EZH2* and a higher coverage on *WT1*. For this sample, BiTSC$^2$ inferred many CNAs which were not validated by SNP array data (Fig. 4C), which are likely due to BiTSC$^2$ not accounting for the uneven coverage in different regions. In addition, COMPASS placed the two different *RUNX1* mutations in different subclones, which is in accordance with the fact that very few cells had both mutations (Fig. 4F, top-right), probably corresponding to doublets. However, BiTSC$^2$ and SCITE placed these two mutations in the same lineage (Fig. 4C, D). For SCITE, this is because the *RUNX1* p.D198N mutation is present in a minority of cells and SCITE assumes that all nodes have the same prior attachment probability, which results in a higher likelihood when the *RUNX1* p.D198N mutation is placed below the other *RUNX1* mutation. This highlights the benefits of the more complex model used by COMPASS over SCITE, even in the absence of CNAs. Other examples where COMPASS and SCITE produce different results, even in the absence of CNAs, are provided in Supplementary Figure 21.

The inferred tree for sample AML-99-001 displays a linear evolution and contains a gain of two genes on chromosome 8, as well as a CNLOH of *RUNX1* on chromosome 21 (Fig. 4H), both of which are validated by the bulk SNP array data (Fig. 4G). One germline variant on *RAD21* was covered by the targeted panel, which improves the reliability of the CNA call on chromosome 8, since it is based both on the total coverage in the region as well as on the allelic fraction of the *RAD21* SNP which increases from 1/2 to 2/3 after the gain (Fig. 4K). Interestingly, there are 5 longitudinal samples available for this patient, and we detect the copy number gain on chromosome 8 and the CNLOH on *RUNX1* in all 5 of them, although the CNLOH on *RUNX1* is only present in a small subclone on the fourth and fifth samples (Supplementary Fig. 19).

### New insights in clonal evolution of myeloid neoplasms

After validating that the CNAs detected by COMPASS in single-cell data match the ones detected in bulk sequencing, we set out to investigate new insights that could be gained by this method. An obvious benefit

of detecting CNAs in single-cell data is the ability to infer subclonal events. For example, the subclonal *TP53* deletion inferred by COMPASS in sample AML-59-001 (Fig. 4A) is not detected in bulk data, but might be very relevant for therapy resistance[30]. The detection of sub-clonal *JAK2* CNLOH (Supplementary Fig. 20) also requires single-cell data, since the bulk variant allele frequency cannot distinguish between 5% of the cells being homozygous for the mutation or 10% of the cells being heterozygous, whereas these two different states might be clinically different, as homozygous *JAK2* mutations result in a stronger phenotype[31]. The ordering of mutations and CNAs inferred by COMPASS can also be very enlightening to understand disease evolution. For example, in sample AML-83-002, COMPASS inferred that the trisomy 8 occurred after the *DNMT3A* and *IDH2* mutations, but before the mutations in *FLT3* and *NRAS* (Fig. 5A).

Sample AML-101-001 provides an interesting illustration of the benefits of joint SNV and CNA phylogeny inferred from single-cell data for understanding the evolution of a tumour. This sample contains two different mutations in *TP53* (on the two different allelic copies) and COMPASS inferred two independent deletions on chromosome 17 (Fig. 5B). In the first deletion, all three genes present in the panel on this chromosome (*TP53*, *NF1* and *PPM1D*) were lost, and in the second deletion only *TP53* and *NF1* were lost. Such double *TP53* mutations are not rare in AML, although they are less common than one *TP53* mutation followed by a LOH[32]. Once both *TP53* alleles are mutated we would not expect any additional fitness advantage from losing one copy, whereas here if two deletions on chromosome 17 were independently selected, it seems likely that this deletion drives oncogenesis. A possible explanation is that the fitness advantage provided by these deletions on chromosome 17 does not come from the loss of *TP53*, but rather from the loss of *NF1*. *NF1* codes for the protein neurofibromin, which is a GTPase-activating protein that can accelerate the hydrolysis of RAS-bound GTP into GDP, thus downregulating the RAS pathway. Consequently, a loss of *NF1* could result in an increased activity of the RAS pathway[33], which has been shown to synergise with *TP53* mutations and leads to a dismal outcome in AML[34]. This proposed mechanism would be consistent with the fact that there are two additional clones which also contain mutations upregulating the RAS pathway (mutations in *KRAS* and *PTPN11*). Thus, this would be a case of convergent evolution where there are four co-existing clones with different genotypes, but all of these genotypes have the same consequence on the RAS pathway. In this example, integration of SNVs and CNAs into the phylogeny is critical because based on the coverage information alone, it would not be possible to detect that two different copies of *TP53* were lost independently.

To further validate our method, we applied it to two independently generated targeted scDNAseq cohorts[19,20]. The first cohort consists of four *TP53*-mutated AML patients which were analysed with the Tapestri® platform, before and after venetoclax treatment for seven days[19]. Two patients (AML-CAL-012 and AML-CAL-030) were sequenced with a 127-amplicon panel and the two other patients (AML-CAU-001 and AML-CVC-001) were sequenced using a 312-amplicon panel. In all four patients, COMPASS detected CNAs (Fig. 5C and Supplementary Fig. 21), with in particular loss of *EZH2* in all four cases. The authors of the original publication noted that these four patients experienced an increase in the *TP53*-mutated clone size after venetoclax treatment, which is also reflected in the cell attachments of the trees generated by COMPASS. In addition, the presence of CNAs in COMPASS-inferred trees can help understand the growth of specific clones. Although we did not observe the emergence of new CNAs during this short treatment, a spectacular decrease of the TP53-wt fraction from 46% to 18% in only seven days for sample AML-CAL012 (Fig. 5C) could indicate that the 7q and 12p deletions as well as the *TP53* CNLOH played a role in this evolution. Then, we applied COMPASS to the second cohort, eight *TP53*-mutated MPN samples analysed with the Tapestri® platform[20].

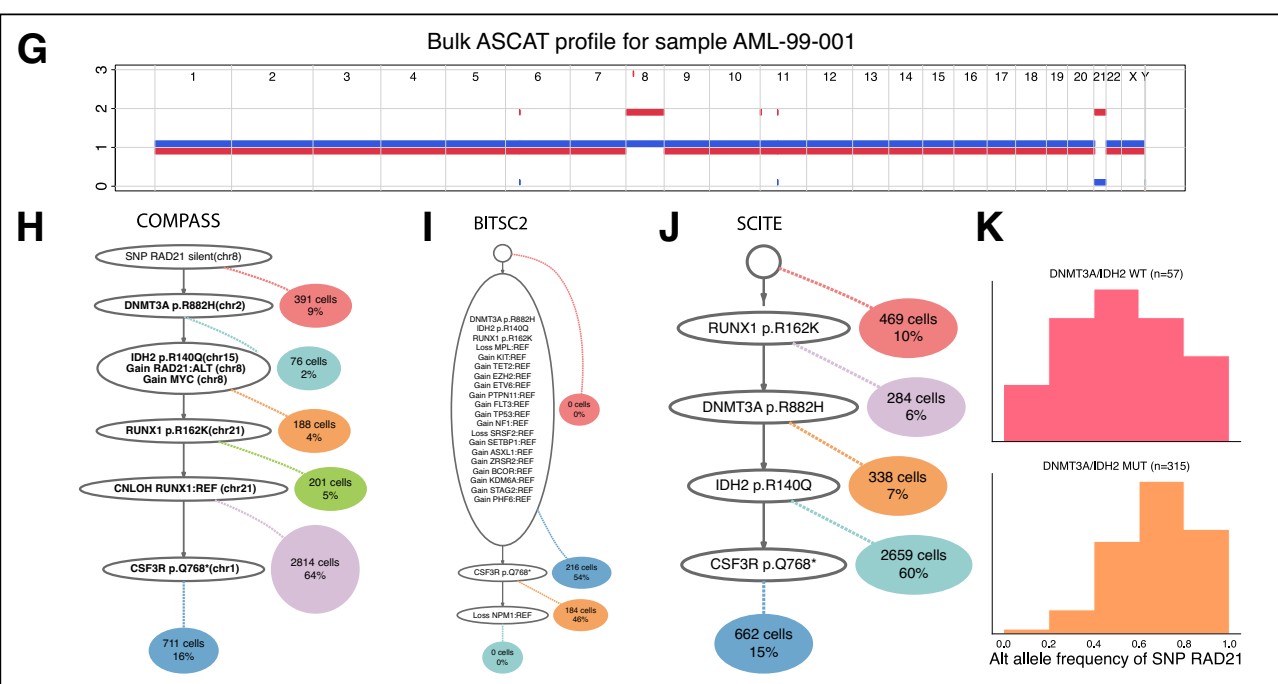

**Fig. 4 | Orthogonal validation of the inferred CNAs with bulk data and comparison with BiTSC² and ∞SCITE. A** ASCAT profile for sample AML-59-001. **B–D** Trees inferred by COMPASS, BiTSC² and ∞SCITE for sample AML-59-001. **E** Boxplots showing the fraction of reads falling on the three genes affected by CNAs in sample AML-59-001, for different genotypes, where *n* indicates the number of cells with each genotype. The boxplot indicate the median and the first and third quartiles, the whiskers show the rest of the distribution up to 1.5 times the interquartile length, and outliers were removed for clarity. **F** Table indicating the number of cells having each possible genotype for the two *RUNX1* mutations in sample AML-59-001. **G** ASCAT profile for sample AML-99-001. **H–J** Trees inferred by COMPASS, BiTSC² and ∞SCITE for sample AML-99-001. **K** Variant allele frequency of the SNP on RAD21 (chr8) for cells with or without the *DNMT3A* and *IDH2* mutations, in sample AML-99-001. *n* indicates the number of cells with each genotype (only cells with a clear genotype for *DNMT3A* and *IDH2* were kept).

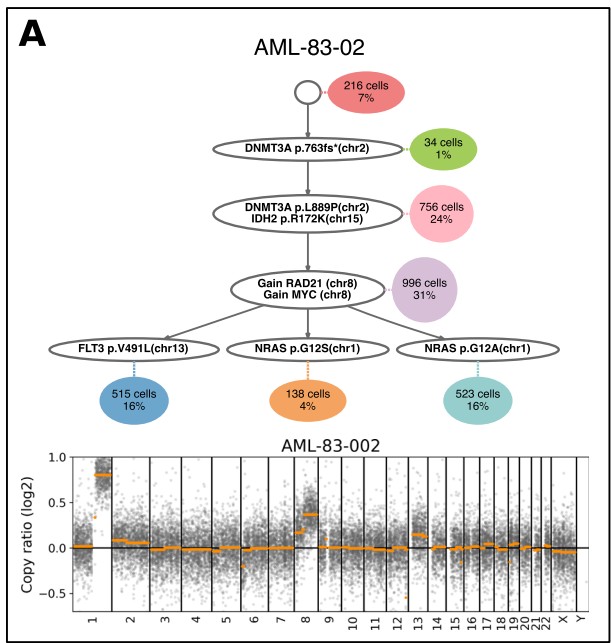

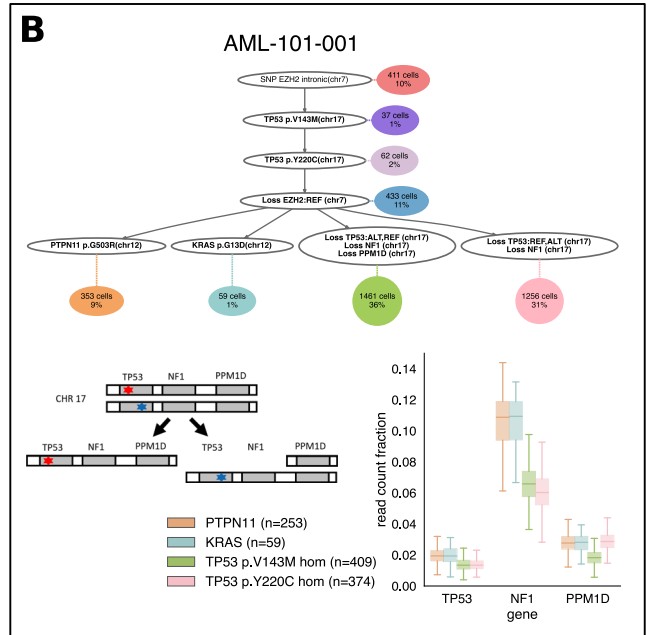

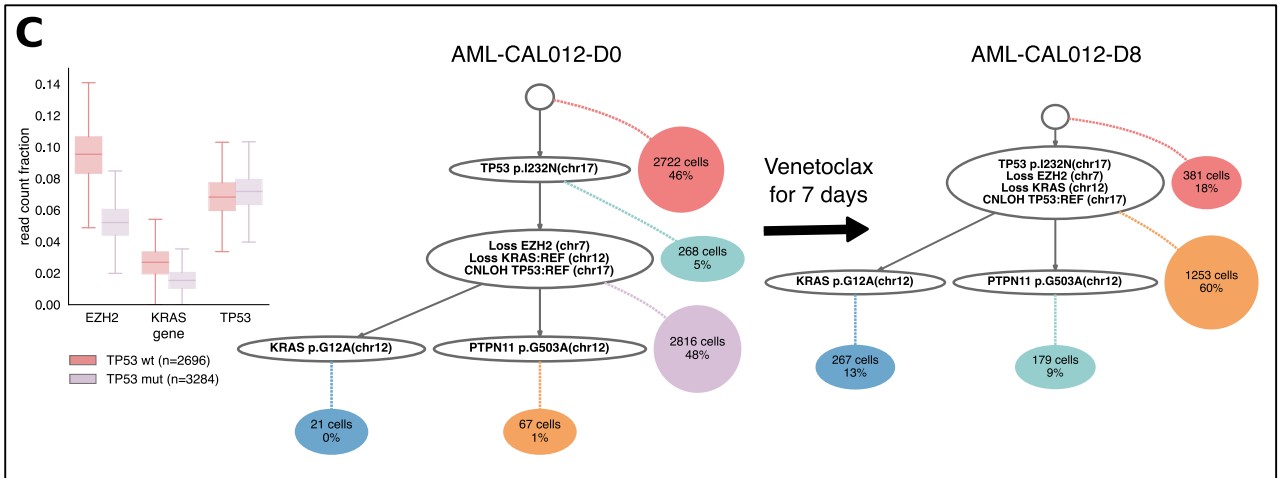

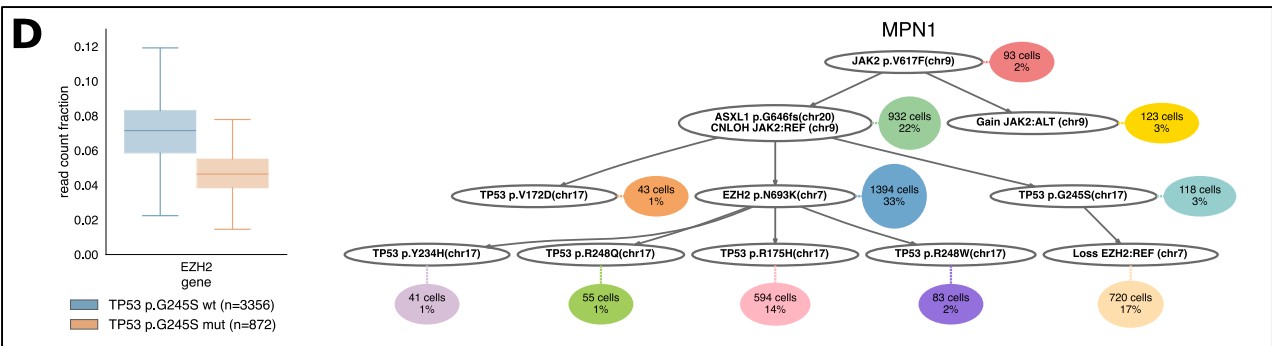

**Fig. 5 | New insights in clonal evolution of myeloid neoplasms. A** Tree inferred by COMPASS for sample AML-83-002, and copy-number plot generated with CNVkit from bulk sequencing. **B** Tree inferred by COMPASS for sample AML-101-001 (top), with a sketch showing the two deletions on chr17 (bottom-left) and the fraction of reads falling on the three targeted genes on chr17 (bottom-right), where *n* indicates the number of cells with each genotype. **C** Trees inferred by COMPASS for sample AML-CAL012, before and after venetoclax treatment, with the fraction of reads in regions affected by CNAs, where *n* indicates the number of cells with each genotype. **D** Tree inferred by COMPASS for sample MPN1, with the sequencing depth on EZH2 for TP53-mut and TP53-wt cells, where *n* indicates the number of cells with each genotype. All boxplots show the median and the first and third quartiles, the whiskers show the rest of the distribution up to 1.5 times the inter-quartile length, and outliers were removed for clarity.

Compared to the trees shown in the original publication, those inferred by COMPASS are generally more compact and do not contain mutation recurrences (Fig. 5D and Supplementary Fig. 23). This is because the presence of cells exhibiting two mutations occurring in different lineages can be explained by COMPASS' doublet model. All eight samples had *JAK2* mutations, and COMPASS detected CNLOH of *JAK2* in three of them (Fig. 5D and Supplementary Fig. 23). Interestingly, COMPASS detected one subclonal deletion of *EZH2* in sample MPN1 (Fig. 5D), and a different subclone harboured an EZH2 p.N693K mutation, which might be a case of convergent evolution where haploinsufficiency of *EZH2* was acquired twice independently, once with a deletion and once with a point mutation. This convergent evolution would have been missed by a phylogeny based only on SNVs. Consequently, COMPASS can help understand MPN evolution, which is crucial since MPN frequently progresses into AML.

## Discussion

We have developed COMPASS, a probabilistic model for inferring clonal phylogenies based on point mutations and copy number events from single-cell DNAseq data. COMPASS is geared towards the use of read count data from high-throughput amplicon-based sequencing, for example, as generated by the MissionBio Tapestri® platform. Unlike BiTSC² which is currently the only other method to infer tumour phylogenies based on SNVs and CNAs from single-cell sequencing data, COMPASS can also detect copy-neutral loss of heterozygosity, an important prognostic marker in AML. Our simulation experiments illustrate two further key advantages of COMPASS over BiTSC². First, COMPASS is able to process larger datasets with thousands of cells, and second, it is more robust to systematic local coverage fluctuations between amplicons that are independent of copy number changes. These patterns are most likely introduced by variability in primer pair efficiency in the targeted amplicon-based sequencing. This feature constitutes a critical improvement over BiTSC², whose read count model assumes a uniform coverage over all loci (in the absence of CNAs). Even though BiTSC² performs well with a uniform coverage, we observed that it generated a large amount of false CNA calls in real amplicon-based data with non-uniform coverage, making it inapplicable to Tapestri® data. In general, SNVs are easier to call from deep-targeted sequencing than copy number states. This stands in contrast to shallow WGS which is better suited for detecting large CNAs than SNVs. Looking at CNA detection alone, our results show that COMPASS has a similar performance as BiTSC² with a uniform coverage, but outcompetes BiTSC² with regard to both false negative and false positive rate with a non-uniform coverage. We also observe that it is particularly challenging to detect subclones characterised only by CNAs, and in our simulations, COMPASS mostly fails to detect them. This is analogous to how CHISEL can only detect SNVs in subclones containing CNAs[15], for shallow WGS data. The same trend is observed for BiTSC² but less pronounced due to the overall less conservative CNA calling strategy of this method. In practice, we still managed to detect subclonal CNAs because these subclones also harboured SNVs. In addition, many of the subclones seemingly characterised only by CNAs will in fact be supported by SNVs located outside the small set of genes currently targeted in high-throughput assays. Therefore sequencing a larger part of the genome will likely reduce the number of these hard to detect CNAs.

We applied COMPASS to a large real-world dataset of 123 AML samples. Previously, clonal architecture of these samples was only inferred based on SNVs. Jointly analysing SNVs and CNAs with COMPASS allowed for a more complete characterisation of the clonal heterogeneity in these samples. The CNAs that were detected by COMPASS are in agreement with current AML knowledge, for example frequent CNLOH of *FLT3* and deletion of the long arm of

chromosome 7, as well as the association between *TP53* mutations and CNAs. In addition, the clonal CNAs could be orthogonally validated by bulk data. The main scientific advance provided by COMPASS is the ability to delineate the order of SNVs and CNAs in a branching evolution pattern, which can help analyse the fitness of clones that are evolving in parallel. In the original publication describing the dataset of 123 AML samples[18], the authors observed several cases of convergent evolution where several subclones had independently acquired mutations leading to a similar phenotype, like one *IDH1* and one *IDH2* mutation in parallel. Here, through a more comprehensive analysis encompassing both SNVs and CNAs, we could detect new cases of convergent evolution that were missed by only considering SNVs, showing that this phenomenon is more prevalent than previously thought. For example, we found cases where a deletion of *NF1* occurred in parallel with mutations in *KRAS* or *PTPN11*, and all these events can lead to the activation of the RAS pathway. Similarly, we found a case of an *EZH2* deletion which occurred in parallel with an *EZH2* mutation.

We applied COMPASS to three independently generated targeted scDNAseq datasets which highlights that the method is applicable to different amplicon panels. While we focus here on AML and MPN, for which a lot of Tapestri® data is currently available, COMPASS is not generally restricted to blood malignancies, although performance on other types of datasets could not be evaluated. More complex copy number events like whole genome duplications which are uncommon in leukemias but occur in many solid tumours could be modelled in COMPASS by adding one event which doubles every copy number.

## Methods

### Probabilistic model

COMPASS defines a probability distribution over trees of somatic events (SNVs and CNAs). The prior on trees penalises the number of nodes and CNAs, and the likelihood takes into account both the total number of reads in each region and the number of reads at each locus supporting the mutated or wild-type alleles. The two components of this likelihood are described in more detail below. A simulated annealing algorithm is used to infer the tree with the highest probability. A complete description of the probabilistic generative process defined for COMPASS is provided in the Supplementary Note 1.

### Likelihood for the number of reads in each region

We observed in targeted sequencing data that the read depth varied a lot from region to region (Supplementary Fig. 6) and for each region, there was a lot of variability across cells (Supplementary Fig. 7).

The non-uniform coverage across amplicons was modelled by giving a weight $\rho_k$ to each region $k$, which represents the probability for a read to fall on this region when there is no CNA. When there are copy number alterations, the probability for a read to fall on region $k$ depends on the copy number of that region, as well as on the copy numbers of the other regions. Thus, if cell $j$ has a total of $D_j$ reads and is attached to node $\sigma_j$ which has a copy number $c_l(\sigma_j)$ for region $l$, then the expected read depth on region $k$ for cell $j$ is $\mathbb{E}(D_{kj}) = D_j \frac{c_k(\sigma_j)\rho_k}{\sum_l c_l(\sigma_j)\rho_l}$.

The high variability of read depth across cells for each amplicon was modelled with a negative binomial distribution (Gamma-Poisson), which is commonly used for modelling read counts[35] and allows for overdispersion. We parameterise the negative binomial distribution with a mean $\mu$ and inverse dispersion parameter $\theta$. This corresponds to sampling the read counts from a Poisson distribution, where the rate of the Poisson distribution is first sampled from a Gamma distribution with shape parameter $\theta$ and scale parameter $\frac{\mu}{\theta}$. Compared to a Poisson distribution whose variance would be equal to the mean $\mu$, the Gamma-Poisson distribution has a higher variance $\mu + \frac{1}{\theta}\mu^2$.

The likelihood for the read depth in region $k$ for cell $j$ is:

$$P(D_{kj}|D_j \frac{c_k(\sigma_j)\rho_k}{\sum_l c_l(\sigma_j)\rho_l}\rho_k, \theta) = \frac{\Gamma(D_{kj}+\theta)}{\Gamma(D_{kj}+1)\Gamma(\theta)}\left(\frac{\theta}{\theta+D_j\frac{c_k(\sigma_j)\rho_k}{\sum_l c_l(\sigma_j)\rho_l}}\right)^\theta$$
$$\left(\frac{D_j\frac{c_k(\sigma_j)}{2}\rho_k}{\theta+D_j\frac{c_k(\sigma_j)\rho_k}{\sum_l c_l(\sigma_j)\rho_l}}\right)^{D_{kj}} \tag{1}$$

### Likelihood for the number of mutated reads at each variable locus

COMPASS does not take as input called genotypes, but instead works directly with the allelic read counts, similar to SCIΦ[9]. Even when the coverage is low, COMPASS can harness all of the available information while taking the uncertainty into account. In addition, a copy number alteration can lead to one allele having a higher copy number than the other, resulting in an unbalanced allelic proportion, which can be detected from the allelic read counts (Supplementary Fig. 11), making the CNA inference more precise. We model allelic read counts with a beta-binomial distribution to account for overdispersion. Let $D$ be the sequencing depth at a position and $A$ be the counts of alternative reads, $f$ be the frequency of the alternative nucleotide and $\omega$ be the concentration parameter. The beta-binomial likelihood is given by

$$P(A|D, f, \omega) = \binom{D}{A}\frac{B(A+\omega f, D-A+\omega(1-f))}{B(\omega f, \omega(1-f))} \tag{2}$$

where $B$ is the beta function.

Let $c^{(r)}$ and $c^{(a)}$ be the number of copies of the reference and alternative allele, respectively. The true proportion of the alternative nucleotide is $\frac{c^{(a)}}{c^{(r)}+c^{(a)}}$. Let $\varepsilon$ be the sequencing error rate. If we exclude the two other nucleotides different from ref and alt, the proportion of alternative reads should be $\frac{c^{(a)}}{c^{(r)}+c^{(a)}}(1-\varepsilon) + \frac{c^{(r)}}{c^{(r)}+c^{(a)}}\varepsilon$. Each of the two alleles can independently be dropped out: Let $k$ and $l$, respectively, be the number of reference and alternative allele copies which got amplified. We observed that in real data, different variants had different dropout rates (Supplementary Figure 10), so we allowed in our model each variant $i$ to have its own dropout rate $\mu_i$, which is inferred using an EM algorithm described below. Taking into account all of the dropout possibilities, the probability of the observed read counts for cell $j$ at locus $i$ is

$$P\left(A_{ij}|D_{ij}, c^{(r)}, c^{(a)}, \mu, \varepsilon, \omega_{\text{hom}}, \omega_{\text{het}}\right)$$
$$= \sum_{\substack{0 \le k \le c^{(r)} \\ 0 \le l \le c^{(a)} \\ (k,l)\ne(0,0)}} \binom{c^{(r)}}{k}\binom{c^{(a)}}{l}\mu_i^{c^{(r)}+c^{(a)}-k-l}(1-\mu_i)^{k+l}$$
$$P\left(A_{ij}|D_{ij}, \frac{l}{k+l}(1-\varepsilon) + \frac{k}{k+l}\varepsilon, \omega(k,l)\right) \tag{3}$$

where $\omega(k,l) = \omega_{\text{hom}}$ if $k=0$ or $l=0$ and $\omega(k,l) = \omega_{\text{het}}$ otherwise (the overdispersion is higher in case of heterozygosity).

### Marginalisation over the attachments of cells to nodes

Instead of sampling the attachments of cells to nodes as part of the MCMC scheme, we compute the likelihood of a tree by marginalising over the attachments and we only sample trees as in SCITE[7]. This makes the inference much faster when the number of cells is high, which is typically the case for Tapestri® data. Unlike SCITE, COMPASS does not use a uniform prior over nodes in the marginalisation, but instead learns the probability $\pi_n$ to sample a cell from a node $n$. This improves the inference, especially when some clones are much smaller

than others, and this is feasible for Tapestri® data because we have a high number of cells and a small number of nodes. The node weights $\pi_n$ are learnt using an EM procedure described below. If $\sigma_j$ denotes the node to which cell $j$ is attached, then the likelihood of a tree can be written as

$$P(\boldsymbol{D}, \boldsymbol{A}|\mathcal{T}) = \prod_{\text{cell } j} \sum_{\sigma_j} \pi_{\sigma_j} \prod_{\text{region } k} P(D_{kj}|c_k(\sigma_j), \rho_k) \prod_{\text{locus } i} P(A_{ij}|D_{ij}, c_i^{(r)}(\sigma_j), c_i^{(a)}(\sigma_j), \mu_i) \tag{4}$$

### Doublets

Doublets can optionally be modelled. In case they are included, we compute separately the probability of a cell to attach to a single node, and to attach to a doublet, and we mix them with the doublet probability $\delta$, as is done in ∞SCITE[8].

The general formula of Eq. (4) remains valid, but the attachment $\sigma_j$ of cell $j$ can either be a single node or a pair of nodes. In case $\sigma_j$ is a single node $n$, the probability to attach to it is $P(\sigma_j) = (1-\delta)\pi_n$. In case $\sigma_j$ is a doublet $(n, n')$, the probability to attach to it is $P(\sigma_j) = \delta\pi_n\pi_{n'}$. The genotype of a doublet is computed by adding the copy numbers of the alleles of the two nodes, and averaging the copy numbers of the regions. If we explicitly separate singlets from doublets, we obtain

$$P(\boldsymbol{D}, \boldsymbol{A}|\mathcal{T}) = \prod_{\text{cell } j}\left((1-\delta)\sum_n \pi_n P(\boldsymbol{D}_j, \boldsymbol{A}_j|n) + \delta\sum_n\sum_{n'}\pi_n\pi_{n'}P(\boldsymbol{D}_j, \boldsymbol{A}_j|n, n')\right) \tag{5}$$

$$P(\boldsymbol{D}, \boldsymbol{A}|\mathcal{T}) = \prod_{\text{cell } j}\left((1-\delta)\sum_n \pi_n \prod_{\text{region } k} P(D_{kj}|c_k(n), \rho_k)\right.$$
$$\prod_{\text{locus } i} P(A_{ij}|D_{ij}, c_i^{(r)}(n), c_i^{(a)}(n), \mu_i)$$
$$+ \delta\sum_n\sum_{n'}\pi_n\pi_{n'}\prod_{\text{region } k} P\left(D_{kj}|\frac{c_k(n)+c_k(n')}{2}, \rho_k\right)$$
$$\left.\prod_{\text{locus } i} P(A_{ij}|D_{ij}, c_i^{(r)}(n)+c_i^{(r)}(n'), c_i^{(a)}(n)+c_i^{(a)}(n'), \mu_i)\right) \tag{6}$$

### Tree prior

The tree prior penalises the number of nodes in the tree, as well as the number of CNA events.

The penalty for the number of nodes is proportional to the number of mutations in the tree because if the tree contains many mutations, it is more likely that each node will contain several mutations.

When several CNA events affect contiguous regions, they are counted as one event, because such events typically affect large genomic regions, often whole chromosomes. When a CNA results in a LOH, it has a stronger impact on the likelihood. Consequently, a stronger penalty is used for CNA events which lead to LOH. The penalty for CNAs has an affine relationship with the number of cells, because when more cells are present, such events have a higher impact on the likelihood, but we also need a minimum evidence to be able to detect such events.

Since COMPASS allows the inclusion of germline variants in the tree (to improve the inference of CNAs, in case they are part of a region affected by a CNA), mutations which are not at the root of the tree are penalised. Optionally, COMPASS can take as input the frequency of variants in the 1000 Genomes database. Variants present in this database are penalised more heavily (proportionally to their population frequency) for not being at the root, since they are more likely to be germline variants.

The prior of a tree depends on parameters for the penalties $p_1$, $p_2$, $p_3$ and $p_4$, which are chosen empirically. The main one is $p_2$ which controls the addition of CNAs. Its default value works well on MissionBio datasets, but might have to be adjusted in case there are too many false positives or false negatives on other datasets. The default value of $p_1$ is low because the node probabilities already remove most of the benefits of having additional nodes, but it could be increased if we wanted to reduce the number of nodes in the tree. The values of $p_3$ and $p_4$ are not critical because this part of the prior does not play a significant role in most cases. The formula for this log-prior is:

$$\log(P(\mathcal{T})) = -p_1 n_{\text{muts}} n_{\text{nodes}}$$
$$- (1500 + n_{\text{cells}}) p_2 \left( n_{\text{CNA\_LOH}} + \frac{1}{2} n_{\text{CNA\_no\_LOH}} \right)$$
$$- \sum_{\text{locus } i} \mathbb{1}_{i \text{ not attached at the root}} \left( p_3 + p_4 \text{freq}_{1000\text{Genomes}}(i) \right) \qquad (7)$$
$$+ \text{Constant}$$

### Simulated annealing

Even though the number of mutations with targeted DNA sequencing is small, the tree space is still very large, which precludes an exhaustive search over the whole tree space. Consequently, we use a simulated annealing (SA) approach. We start from a randomly generated tree. The number of nodes $n_{\text{nodes}}$ of the initial tree is randomly chosen between 3 and 10. We generate a random sequence of $n_{\text{nodes}} - 2$ integers in $[0, n_{\text{nodes}}]$ which we interpret as a Prüfer sequence to assign a parent to each node. The SNVs are randomly assigned to the nodes, and the initial tree does not contain any CNA. Then, at each iteration, we propose a new tree $\mathcal{T}'$ from the current tree $\mathcal{T}$ by sampling it from a proposal distribution $q(\mathcal{T}, \mathcal{T}')$. The MCMC moves are described in the Supplementary Note 2. Then, we compute the likelihood of the new tree, and accept the new tree with probability $\min\left\{ 1, \exp\left( \frac{\log(P(\mathcal{T})P(D|\mathcal{T}))}{T} \right) \right\}$ where $T$ is a temperature parameter. Otherwise, we reject the new tree and start a new iteration from tree $\mathcal{T}$. The temperature is progressively lowered, which prevents being stuck in a local optimum initially.

In practice, we first run SA without CNAs. That way, we can identify the cells that are attached to the root as non-neoplastic cells, and use those cells to estimate the weight of each region $\rho_k$, which is the probability for a read to fall into region $k$ for a diploid cell without any CNAs. In addition, in the inferred tree without CNAs, we look for regions which have a lower or higher average normalised sequencing depth in some nodes compared to the root, and we select those regions as potential regions which might harbour copy number variants. Then, we run the SA with CNAs, but we restrict the addition of CNA events to the selected regions. We also exclude regions which have a very low amplification rate from the CNA inference, as their sequencing depth is very unreliable. This selection might lead to false negative CNAs, but reduces the number of false positives and decreases the number of iterations required in the SA, since it reduces the set of possible events that can be proposed.

### Estimation of the node probabilities and dropout rates

The model contains two parameters which need to be estimated: the weight $\pi_n$ of each node $n$ and the dropout rate $\mu_i$ of each variant $i$. Ideally, we would like to marginalise over these parameters. However, the space is too large to integrate over, and sampling these parameters with the MCMC would be very inefficient: when a new tree is proposed, the old parameters might not work well for this new tree, which would lead to the tree being refused with a very high probability. Alternatively, we could jointly propose a new tree and new node weights and dropout rates, but the probability to obtain good parameters would be extremely low.

Thus, instead of marginalising over the node probabilities and dropout rates, we use the parameters which maximise the posterior probability. This can be efficiently performed with an EM algorithm, which has to be performed inside each MCMC step. We have two types of latent variables: the attachments of cells to nodes, $\sigma_j$, and for each cell $j$ and each locus $i$, the number of reference and alternative alleles that did not get dropped out, $C_{ij}^{(r)}$ and $C_{ij}^{(a)}$. We use a beta prior centred on 0.05 for the dropout rates and a flat Dirichlet prior $D(1, ..., 1)$ for the node weights.

During the E-step, we compute the probabilities $Q$ of the latent variables, given the current parameters.

$$Q(\sigma_j = n) = P(\sigma_j = n | \mathbf{D}_j, \mathbf{A}_j, \boldsymbol{\pi}, \boldsymbol{\mu}) = \frac{\pi_n P(\mathbf{D}_j, \mathbf{A}_j | \sigma_j = n, \boldsymbol{\pi}, \boldsymbol{\mu})}{\sum_{n'} \pi_{n'} P(\mathbf{D}_j, \mathbf{A}_j | \sigma_j = n', \boldsymbol{\pi}, \boldsymbol{\mu})} \qquad (8)$$

$$Q(C_{ij}^{(r)} = k, C_{ij}^{(a)} = l | \sigma_j = n) = P(C_{ij}^{(r)} = k, C_{ij}^{(a)} = l | D_{ij}, A_{ij}, \sigma_j = n, \boldsymbol{\mu})$$
$$= \frac{\mu_i^{c_i^{(r)}(n) + c_i^{(a)}(n) - k - l} (1 - \mu_i)^{k+l} P(A_{ij} | D_{ij}, k, l)}{\sum_{k', l} \mu_i^{c_i^{(r)}(n) + c_i^{(a)}(n) - k' - l'} (1 - \mu_i)^{k' + l'} P(A_{ij} | D_{ij}, k, l)} \qquad (9)$$

During the M-step, we update the parameters (node probabilities $\pi_n$ and dropout rates $\mu_i$) in order to maximise the sum of the log-prior and of the expected hidden log-likelihood.

$$\pi_n = \frac{1}{n_{\text{cells}}} \sum_{\text{cell } j} Q(\sigma_j = n) \qquad (10)$$

$$\mu_i =$$
$$\frac{\alpha - 1 + \sum_{\text{node } n} \sum_{\text{cell } j} Q(\sigma_j = n) \sum_{k,l} Q\left( C_{ij}^{(r)} = k, C_{ij}^{(a)} = l | \sigma_j = n \right) \left( c_i^{(r)}(n) + c_i^{(a)}(n) - k - l \right)}{\alpha + \beta - 2 + \sum_{\text{node } n} \sum_{\text{cell } j} Q(\sigma_j = n) \left( c_i^{(r)}(n) + c_i^{(a)}(n) \right)}$$
$$(11)$$

### Reporting summary

Further information on research design is available in the Nature Portfolio Reporting Summary linked to this article.

## Data availability

No new data was generated in this study, but we used scDNAseq data that had been previously generated for three different publications using the Tapestri® platform from MissionBio. The data of Morita et al.[18] is available on the SRA under the project ID PRJNA648656 and the authors shared the processed loom files with us. The raw data for the TP53-mutated AML samples treated with venetoclax[19] and for the TP53-mutated MPN samples[20] are not publicly available, but the authors kindly shared the loom files with us. For all three datasets, the pre-processed data that is used as input for COMPASS is available on GitHub at https://github.com/cbg-ethz/COMPASS. For figures, source data are provided as a Source Data file. Source data are provided with this paper.

## Code availability

COMPASS has been implemented in C++ is freely available under a GPL3 license at https://github.com/cbg-ethz/COMPASS. The repository also contains preprocessing python scripts, as well as the snakemake pipeline that was used to run all simulations.

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

## Acknowledgements

This work was partially supported by SNSF Grant 310030 179518 (http://www.snf.ch, to N.B. and J.K.), as well as the AML/MDS Moonshot Grant from MD Anderson (to K.T.) and the Leukemia Lymphoma Society Scholar Award (to K.T.). We thank Fiona Brown (Australian Centre for Blood Diseases, Monash University, Melbourne, Australia) for sharing the loom files for the MissionBio dataset of TP53-mutated AML treated with venetoclax, and Nabih Maslah, Bruno Cassinat and Jean-Jacques Kiladjian (Assistance Publique - Hôpitaux de Paris; Université de Paris, INSERM, Institut de recherche Saint-Louis, Paris, France) for sharing the loom files for their MissionBio data for TP53-mutated MPN.

## Author contributions

J.K. and K.J. conceptualised the study. E.S., J.K. and K.J. designed the methodology. E.S. implemented the methodology. E.S. and K.T. performed the biological analysis. J.K., N.B. and K.J. provided supervision. E.S. wrote the original draft. All authors reviewed and approved the paper.

## Funding

## Competing interests

The authors declare no competing interests.
