## [Peer Review File · Nature Communications]

COMPASS: Joint copy number and mutation phylogeny reconstruction from amplicon single-cell sequencing dataREVIEWER COMMENTS

Reviewer #1 (Remarks to the Author):

Sollier et al presented a computational method COMPASS that can infer tumor cell phylogeny based on joint copy number variations (CNV) and single nucleotide variations (SNV) from single-cell amplicon sequencing data. Using both simulated data and a published AML dataset, the authors demonstrated that COMPASS can outperform existing methods such as BiTSC2 and SCITE. Overall, the presentation of the work is quite clear. However, joint CNV and SNV inference of tumor phylogeny has been explored in several recent methods including BiTSC2. Comparing to earlier methods, COMPASS' improvements are quite technical and the re-analysis of the AML data does not yield significant novel insights, my major concern is lacking of significant novelty in the current work.

1) The authors claimed that one of the major improvements of COMPASS is modeling CNLOH events. However, it is not clear how the authors generate CNLOH in the simulation and whether COMPASS allows multiple hits of SNVs or CNVs in the same region.

2) The hyperparameters for tree priors p1-p4 have to be chosen empirically and seem to be tuned specifically for AML in the presentation. Since liquid tumors can be quite different from solid tumors, it is unknown whether extensive prior knowledge is required for the users to pick these parameters?

3) The analysis of the AML dataset yielded many results. It is hard to gauge what is novel both methodologically and biologically.

a) What findings are unique to COMPASS that is not identifiable in other methods?

b) Biologically, what novel insights the new analysis tell us about AML evolution? I found many of these findings are known earlier. It will be important to highlight the novelty.

4) Several points presented in the work is not accurate:

a) The authors claimed that " BiTSC2 does not allow CNLOH events and might therefore falsely interpret them as copy number losses. In addition, it does not allow losses of the mutated allele and only allows a copy-number gain of the mutated allele when it occurs in the same node as the corresponding SNV." This is inaccurate. BiTSC2 do account for loss of mutation. In fact, the phase indicator in the original paper's fig1d has shown that both copy gain and loss are allowed to occur on the mutated allele in BiTSC2.

b) In addition, the authors claim that "BiTSC2 only uses the coverage at SNVs to detect CNVs, which might miss CNVs in regions without SNVs". BiTSC2 also considers genomic regions with only CNA events (Fig1B in the paper). BiTSC2 offers to jointly infer CN event in consecutive regions within one segment, when segmentation info is available.

c) The abstract of the work is written as if COMPASS is the first method jointly modeling CNV and SNV. This is not true. In other words, the "gap in the field" is not properly described.

Reviewer #2 (Remarks to the Author):

This paper introduces COMPASS, a tumor phylogeny inference method of SNVs and CNAs from single-cell panel sequencing data. The focus is on MissionBio Tapestry sequencing technology. CNA states include loss, gain and copy-neutral loss of heterozygosity and SNVs are phased with CNAs, i.e. allele-specific CNAs. The method is evaluated on real and simulated data. As this is the first method to simultaneously infer CNAs and SNVs in a phylogeny, I am positive about the contribution of the paper. However, I have several concern that I will discuss below.

1. Experimental evaluation

Simulations should be extended to assess:

- Assess cell attachment proportions w.r.t. to ground truth.
- MP3 similarity combines SNV and CNA accuracy. Please separate these out. There seems to be an imbalance in number of SNVs and CNAs -- so the MP3 score might be dominated by SNVs.
- Is the mutation copy-number of SNVs inferred correctly -- $c_i^{\{r\}}(n)$?
- Can subclonal CNA events be inferred?
- How well can the method infer integer copy-numbers beyond gain/loss?

I have the following comments about the real data:

- How did the competing methods (SCITE and BiTSC2) perform on real data?

2. Methods

- My biggest concern is that a description of the used evolutionary model is missing. Can an SNV be introduced multiple times on the tree? Can an SNV be lost after introduction, if so, how many times. Can the same region be affected by multiple CNAs? Please make this clear.

Relatedly, Section B needs to be written more carefully. Several definitions are missing. In particular, the definition of an event is missing.

- My other concern is that the algorithm consists of many hardcoded values, including the doublet rate, several overdispersion parameters, the allele-specific ADO rate, etc. I would like to understand how these values were chosen, and if they need to be altered depending on the data?

One particular example is λ , which was set to e^{-70} . Does it ever happen that the algorithm uses different allelic dropout rates? Why not just remove this?

- I do not like the separation of CNV (which include gains and losses) and CNLOH, a copy-neutral loss of heterozygosity is a type of CNA in addition to copy-number gains and losses.

- Section "Likelihood for the number of reads" needs more motivation. First, it is unclear why there is a factor of $1/2$ in $E[D_{\{kj\}}]$. Second, the number of reads of a region does not only depend on the copy number of the region itself but also the copy-number of other regions. The larger the copy numbers of other regions are, the smaller the number of reads of the region in question.

- Are ρ_k values consistent across patients that use the same set of amplicons? This is another form of orthogonal validation. Alternatively, should these be inferred a priori using a cohort level analysis?

- Section A.2 describes a generative process, but the starting point is unclear. It seems that you assume that a tree with N nodes with reference and variant copies for each locus has been drawn already. Please describe this more carefully.

- How is the SA algorithm initialized? That is, how are initial trees generated?

- λ was set to e^{-70} . Does it ever happen that the algorithm uses different allelic dropout rates? Why not just remove this?

- The plate diagram in Figure A.1 indicates that $A_{\{ij\}}$ depends on $D_{\{kj\}}$. Should this be $D_{\{ij\}}$ instead?

3. Data availability

As far as I can tell, The repository contains only one simulated instance and a handful of real data instances. Please upload all simulated and processed real data to facilitate reproducibility.

RESPONSE TO REVIEWERS' COMMENTS

We are grateful to the reviewers for their careful reading of our paper and their thoughtful suggestions. We have addressed all reviewer comments and adapted our manuscript accordingly.

In the following, reviewer comments are in black font, our responses in blue.

Reviewer #1 (Remarks to the Author):

Sollier et al presented a computational method COMPASS that can infer tumor cell phylogeny based on joint copy number variations (CNV) and single nucleotide variations (SNV) from single-cell amplicon sequencing data. Using both simulated data and a published AML dataset, the authors demonstrated that COMPASS can outperform existing methods such as BiTSC2 and SCITE. Overall, the presentation of the work is quite clear. However, joint CNV and SNV inference of tumor phylogeny has been explored in several recent methods including BiTSC2. Comparing to earlier methods, COMPASS' improvements are quite technical and the re-analysis of the AML data does not yield significant novel insights, my major concern is lacking of significant novelty in the current work.

The improvements of COMPASS over BiTSC2 are not just technical, because BITSC2 is simply not applicable to amplicon sequencing data, for two main reasons:

- Unlike BiTSC2, COMPASS allows different coverages for each region, which is necessary for targeted sequencing data
- COMPASS can handle large datasets with >10,000 cells, whereas BITSC2 is limited to ~100 cells. This is due to the fact that COMPASS marginalizes over the cell attachments, whereas BiTSC2 samples the attachments.

1) The authors claimed that one of the major improvements of COMPASS is modeling CNLOH events. However, it is not clear how the authors generate CNLOH in the simulation and whether COMPASS allows multiple hits of SNVs or CNVs in the same region.

The synthetic data generation is described in Supplementary Section D1. SNVs can occur only once. CNAs (including CNLOH) can occur multiple times, but at most once per lineage. We added a few sentences in the first paragraph of the results section and modified Figure 1 to make this more explicit.

2) The hyperparameters for tree priors p_1 - p_4 have to be chosen empirically and seem to be tuned specifically for AML in the presentation. Since liquid tumors can be quite different from

solid tumors, it is unknown whether extensive prior knowledge is required for the users to pick these parameters?

The hyperparameters have indeed to be chosen empirically, but they work well on both real and simulated data. COMPASS contains many parameters, so we feel that it's safer to fix some of them than to risk overfitting with too many parameters. The parameters worked on different independently generated datasets that use different amplicon panels, so we expect them to be fairly robust. If necessary, they can be adapted by the user, and we describe on GitHub how to change them.

3) The analysis of the AML dataset yielded many results. It is hard to gauge what is novel both methodologically and biologically.

a) What findings are unique to COMPASS that is not identifiable in other methods?

Subclonal CNAs can only be detected by COMPASS, which provides information about which CNAs co-occur or are mutually exclusive with SNVs in different clones. COMPASS can also provide information on the relative order of acquisition of SNVs and CNAs. In principle, BiTSC2 can also do this, but as we showed in the manuscript, BiTSC2 calls too many false CNAs for amplicon sequencing to be usable in this context.

b) Biologically, what novel insights the new analysis tell us about AML evolution? I found many of these findings are known earlier. It will be important to highlight the novelty.

We adapted the manuscript to make the distinction clearer between confirmation of existing knowledge in Figure 3 and new insights brought by COMPASS in Figure 4. For example, we detected cases of convergent evolution which would have been missed if CNAs were not taken into account, like an EZH2 mutation in one subclone and an EZH2 deletion in a different subclone (Figure 4D).

4) Several points presented in the work is not accurate:

a) The authors claimed that “BiTSC2 does not allow CNLOH events and might therefore falsely interpret them as copy number losses. In addition, it does not allow losses of the mutated allele and only allows a copy-number gain of the mutated allele when it occurs in the same node as the corresponding SNV.”. This is inaccurate. BiTSC2 do account for loss of mutation. In fact, the phase indicator in the original paper's fig1d has shown that both copy gain and loss are allowed to occur on the mutated allele in BiTSC2.

We are sorry for this inaccuracy in our manuscript. When the COMPASS manuscript was originally written, BiTSC2 was not published in Briefings in Bioinformatics yet, and we were referring the preprint which had been uploaded to Biorxiv on 2nd December 2020 (<https://www.biorxiv.org/content/10.1101/2020.11.30.380949v1>). In that earlier version of BiTSC2, the “phase indicator” g was not present, and it was not possible to lose a mutation. When BiTSC2 was published, we updated the reference, but did not notice the addition of the “phase indicator”. We have now updated the COMPASS manuscript accordingly, and we have

rerun all simulations with the new version of BiTSC2. BiTSC2's results did not change noticeably with the new version.

b) In addition, the authors claim that "BiTSC2 only uses the coverage at SNVs to detect CNVs, which might miss CNVs in regions without SNVs". BiTSC2 also considers genomic regions with only CNA events (Fig1B in the paper). BiTSC2 offers to jointly infer CN event in consecutive regions within one segment, when segmentation info is available.

BiTSC2 can indeed use segments to lose or gain one whole segment at once, but as far as we understand BiTSC2, the segments correspond to a collection of SNVs. If one region does not contain any SNV, then there will be no corresponding segment. Figure 1B of the BiTSC2 manuscript does indeed suggest that it is possible to have a gain without any mutation, but this is not possible in the version of BiTSC2 which is currently available on GitHub. Indeed, in this version, all SNVs occur exactly once, so it is not possible to have one locus with a CNA without SNV. We contacted the authors of the BiTSC2 manuscript, and they suggested to add one locus for each region without SNV, and to set the number of alternative reads to 0. All of these "false SNVs" are then placed by BiTSC2 in one node where no cells are attached, and the authors of BiTSC2 recommended that we remove this additional node. This is what we did in the updated version of the manuscript, in spite of the significant amount of pre- and post-processing that this required, and we rerun all simulations with this new version of BiTSC2. This improved BiTSC2's performance when the region coverage was uniform, since BiTSC2 was now able to detect all CNAs. However, when the coverage was more variable across regions, this led to more false positive calls for BiTSC2, since BiTSC2 could now call CNAs on more loci.

c) The abstract of the work is written as if COMPASS is the first method jointly modeling CNV and SNV. This is not true. In other words, the "gap in the field" is not properly described.

We rewrote the abstract accordingly. However, we maintain that, although BiTSC2 is already one method performing a joint SNV and CNV phylogeny inference, it is not applicable to targeted sequencing where different amplicons have different coverages. Thus, COMPASS remains the first method that can infer a joint SNV and CNV phylogeny from targeted scDNAseq data.

Reviewer #2 (Remarks to the Author):

This paper introduces COMPASS, a tumor phylogeny inference method of SNVs and CNAs from single-cell panel sequencing data. The focus is on MissionBio Tapestry sequencing technology. CNA states include loss, gain and copy-neutral loss of heterozygosity and SNVs are phased with CNAs, i.e. allele-specific CNAs. The method is evaluated on real and simulated data. As this is the first method to simultaneously infer CNAs and SNVs in a phylogeny, I am positive about the contribution of the paper. However, I have several concerns that I will discuss below.

1. Experimental evaluation

Simulations should be extended to assess:

- Assess cell attachment proportions w.r.t. to ground truth.

We have now added this evaluation, in Figure 2.

- MP3 similarity combines SNV and CNA accuracy. Please separate these out. There seems to be an imbalance in number of SNVs and CNAs -- so the MP3 score might be dominated by SNVs.

We now provide in Figure 2 different evaluation metrics: the full MP3 similarity, the MP3 similarity restricted to SNVs and the MP3 similarity restricted to CNAs.

- Is the mutation copy-number of SNVs inferred correctly -- $c_i^{(r)}(n)$?

This is evaluated as part of the MP3 similarity.

- Can subclonal CNA events be inferred?

Subclonal CNA events can be inferred, but in general they are only detected by COMPASS when they are supported by SNVs. A CNA is supported by an SNV if it is located in a node which contains a LOH event or an SNV, or one of its descendants contains such an event. This is because, since read depth is quite noisy in single-cell data, subclonal CNAs not supported by SNVs would not be reliable. We added a sketch in Figure 2D to make the definition of CNAs supported by SNVs clear.

- How well can the method infer integer copy-numbers beyond gain/loss?

The method does not infer integer copy numbers but only gains and losses. Considering how noisy single-cell data is, integer copy numbers would not be reliable. BITSC2 estimates integer copy number states, but does not infer them correctly when the coverage is not uniform.

I have the following comments about the real data:

- How did the competing methods (SCITE and BiTSC2) perform on real data?

Figure 3 now contains trees inferred by SCITE and BiTSC2 on real data. BiTSC2 inferred a large number of likely false CNAs (since BiTSC2 inferred these CNAs to be present in all of the cells, but these CNAs were not detected in bulk data). In general, if a region has a coverage higher than the mean, BiTSC2 will infer a clonal copy number gain, and likewise for deletions. Apart from CNAs, the trees inferred by SCITE and COMPASS are similar, except in the cases that we described in Supplementary Section F4.

2. Methods

- My biggest concern is that a description of the used evolutionary model is missing. Can an SNV be introduced multiple times on the tree? Can an SNV be lost after introduction, if so, how many times. Can the same region be affected by multiple CNAs? Please make this clear.

SNVs occur exactly once. CNAs can occur multiple times, but at most once per lineage (as mentioned before, COMPASS does not infer integer copy numbers, so it cannot account for multiple successive gains in the same region, and a loss followed by a gain or vice-versa would be very challenging to detect in noisy amplicon data). Consequently, SNVs can be lost multiple times in different lineages (Dollo model). We added two sentences in the first paragraph of the results section, and added schematics to Figure 1 to make this more explicit.

Relatedly, Section B needs to be written more carefully. Several definitions are missing. In particular, the definition of an event is missing.

We rewrote this section and precisely defined events in Figure 1A.

- My other concern is that the algorithm consists of many hardcoded values, including the doublet rate, several overdispersion parameters, the allele-specific ADO rate, etc. I would like to understand how these values were chosen, and if they need to be altered depending on the data? One particular example is λ , which was set to e^{-70} . Does it ever happen that the algorithm uses different allelic dropout rates? Why not just remove this?

The algorithm contains many parameters, but is generally not very sensitive to most of them. Inferring all of them would be complicated and risk overfitting, which is why we set some of them to hard-coded values. We used values which worked well across different datasets, and we expect them to be robust across most datasets. In the GitHub repository, we briefly defined these parameters and how one could adapt them if necessary.

It sometimes happens that the two alleles have different dropout rates (see Supplementary Figure C4). Since the number of cells in Tapestry datasets is very high, we used a very low value for λ so that it is not negligible compared to the likelihood of thousands of cells.

- I do not like the separation of CNV (which include gains and losses) and CNLOH, a copy-neutral loss of heterozygosity is a type of CNA in addition to copy-number gains and losses.

We rewrote the manuscript by considering CNLOH as a type of CNA. So now CNAs can be a gain, a loss, or a CNLOH.

- Section "Likelihood for the number of reads" needs more motivation. First, it is unclear why there is a factor of $1/2$ in $E[D_{\{kj\}}]$. Second, the number of reads of a region does not only depend on the copy number of the region itself but also the copy-number of other regions. The larger the copy numbers of other regions are, the smaller the number of reads of the region in question.

It is true that the number of reads in a region does not only depend on the copy number of that region, but also on the copy numbers of other regions. However, normalizing by the copy numbers of all regions implies that each time there is a CNA in one node, the likelihoods for the read depth in each region have to be recomputed, which is computationally expensive. In the original version of COMPASS, we omitted this normalization. Thus, we use a factor of $1/2$ which assumed that the genome was diploid. This had the advantage of making the read depth likelihoods in each region independent. Consequently, for each node, we only needed to re-compute the likelihoods for the regions which were affected by a CNA. In most situations, the number of CNAs is either small or the gains and losses balance each other out, making the approximation reasonable. However, this assumption may not be valid for every tumor. Thus, we updated COMPASS to now normalize by the copy numbers of all regions. This makes the algorithm a bit slower, but the runtime remains reasonable (as shown in Figure 2).

We also rewrote the section "Likelihood for the number of reads" in a way that should hopefully make the motivation clearer.

- Are ρ_k values consistent across patients that use the same set of amplicons? This is another form of orthogonal validation. Alternatively, should these be inferred a priori using a cohort level analysis?

The ρ_k values are generally consistent across patients (Supplementary Figure C.3). In principle, it is possible to do a cohort-level analysis, but we wanted to provide to the end-users the option to run COMPASS on one sample only. However, we added one option to give as input to COMPASS the ρ_k values, which can, for example, be estimated with several samples without CNAs.

- Section A.2 describes a generative process, but the starting point is unclear. It seems that you assume that a tree with N nodes with reference and variant copies for each locus has been drawn already. Please describe this more carefully.

We indeed assume that a tree is already drawn. We added a sentence at the beginning of Supplementary Section A2 to make it more explicit. We do not have a clear generative process, but rather use an empirical prior which serves as a penalty for more complex trees.

- How is the SA algorithm initialized? That is, how are initial trees generated?

We apologize for not mentioning this initialization. We have now added the following sentences to the simulated annealing paragraph in the methods section: "We start from a randomly generated tree. The number of nodes n_{nodes} of the initial tree is randomly chosen between 3 and 10. We generate a random sequence of $n_{\text{nodes}}-2$ integers in $[0, n_{\text{nodes}}]$ which we interpret as a Prüfer sequence to assign a parent to each node. The SNVs are randomly assigned to the nodes, and the initial tree does not contain any CNA"

- Lambda was set to e^{-70} . Does it ever happen that the algorithm uses different allelic dropout rates? Why not just remove this?

Yes it does happen in some real datasets, although rarely. MissionBio Tapestry datasets contain thousands of cells, hence the values for the negative log-likelihoods can be very high, which is why this parameter has a high negative log-likelihood.

- The plate diagram in Figure A.1 indicates that $A_{\{ij\}}$ depends on $D_{\{kj\}}$. Should this be $D_{\{ij\}}$ instead?

Yes, this should indeed be $D_{\{ij\}}$. Thank you! We adapted the diagram accordingly.

3. Data availability

As far as I can tell, The repository contains only one simulated instance and a handful of real data instances. Please upload all simulated and processed real data to facilitate reproducibility.

We uploaded the processed real data to the GitHub repository (<https://github.com/cbg-ethz/COMPASS/tree/master/data>). The GitHub repository contains scripts to generate the simulated data.

REVIEWER COMMENTS

Reviewer #1 (Remarks to the Author):

The authors are extremely confident in their findings and the reply to the reviewers' comments. Even though target amplicon sequencing is a new development in the field, it is not yet a main-stream approach. The authors did make significant progress comparing to previous methods such as BiTSC2 on the amplicon sequencing data. However, the simulations were constructed in such a way that those settings are extremely friendly to their own method, while disfavoring previous approaches (e.g. BiTSC2). I feel this is not fair to the colleagues who developed those methods earlier and can mislead the readers. In addition, the authors highly tuned their method on the AML dataset, which makes it hard to know the generality of these performances on other real datasets.

Reviewer #2 (Remarks to the Author):

The revised manuscript has addressed my previous concerns.

RESPONSE TO REVIEWERS' COMMENTS

We are grateful to the reviewers for their careful reading of the revised manuscript and their thoughtful suggestions. We have addressed all reviewer comments and adapted our manuscript accordingly.

In the following, reviewer comments are in black font, our responses in blue.

Reviewer #1 (Remarks to the Author):

The authors are extremely confident in their findings and the reply to the reviewers' comments. Even though target amplicon sequencing is a new development in the field, it is not yet a main-stream approach. The authors did make significant progress comparing to previous methods such as BiTSC2 on the amplicon sequencing data. However, the simulations were constructed in such a way that those settings are extremely friendly to their own method, while disfavoring previous approaches (e.g. BiTSC2). I feel this is not fair to the colleagues who developed those methods earlier and can mislead the readers. In addition, the authors highly tuned their method on the AML dataset, which makes it hard to know the generality of these performances on other real datasets.

The simulation settings were designed to be as close as possible to targeted single-cell DNaseq data, for which our method is specifically designed. We believe that targeted single-cell DNaseq will likely be used much more often in the future due to its cost efficiency. One important difference of targeted scDNaseq as compared to whole-genome scDNaseq is that the coverage is not uniform, while other methods like BiTSC2 assume that the coverage is uniform. The non-uniformity of the coverage has a very strong impact on the performance of BiTSC2 and with these settings, BiTSC2 performed poorly (Figure 2). When we applied BiTSC2 to real targeted sequencing data, the trees generated by BiTSC2 contained many false CNAs which were not detected in bulk data (Figure 3), so we believe that our simulations accurately reflect true targeted sequencing data. Since in our previous version we only focused on targeted sequencing data, it is true that the previous version of the manuscript give the misleading impression that BiTSC2 performs poorly in all settings, even with a uniform coverage. In order to provide a fairer comparison and a more complete picture, we have now separated Figure 2 into two main simulations: one where the coverage is uniform, which acknowledges that BiTSC2 performs well in this setting, and one with non-uniform coverage, which shows the advancement of COMPASS in this setting.

Concerning the applicability of our method to other datasets, we have already in the previous revision added two additional datasets. They are admittedly still from blood malignancies because to our knowledge no similar datasets from solid tumors have been published so far, but

they show that the method works well for other datasets generated in different labs and with different amplicons.

Reviewer #2 (Remarks to the Author):

The revised manuscript has addressed my previous concerns.